# Influence of El Niño-Southern Oscillation regimes on East African vegetation and its future implications under RCP 8.5 warming scenario

Istem Fer[1,2,3], Britta Tietjen[4,5], Florian Jeltsch[1,5], and Christian Wolff[6,7]

[1]Department of Plant Ecology and Nature Conservation, Institute of Biochemistry and Biology, University of Potsdam, Am Mühlenberg 3, 14476 Potsdam, Germany.
[2]DFG Graduate School, Shaping the Earth's Surface in a Variable Environment, University of Potsdam, Karl-Liebknecht-Str. 24, 14476 Potsdam, Germany.
[3]Department of Earth and Environment, Boston University, 685 Commonwealth Ave, 02215 MA, USA.
[4]Biodiversity and Ecological Modelling, Institute of Biology, Freie Universität Berlin, Altensteinstr. 6, 14195 Berlin, Germany.
[5]Berlin-Brandenburg, Institute of Advanced Biodiversity Research (BBIB), D-14195 Berlin, Germany.
[6]Climate Geochemistry Department, Max-Planck Institute for Chemistry, Hahn-Meitner Weg 1, 55128 Mainz, Germany.
[7]International Pacific Research Center, School of Ocean and Earth Science and Technology, University of Hawai'i at Manoa, Honolulu, 96822 HI, USA.

*Correspondence to:* Istem Fer (fer.istem@gmail.com)

**Abstract.** The El Niño Southern Oscillation (ENSO), is the main driver for the interannual variability in East African rainfall with significant impact on vegetation and agriculture, and dire consequences for food and social security. In this study, we identify and quantify the ENSO contribution to the East African rainfall variability to forecast future East African vegetation response to rainfall variability related to a predicted intensified ENSO. To differentiate the vegetation variability due to ENSO, we removed the ENSO signal from the climate data using Empirical Orthogonal Teleconnections (EOT) analysis. Then, we simulated the ecosystem carbon and water fluxes under the historical climate without components related to ENSO teleconnections. We found ENSO driven patterns in vegetation response and confirmed that EOT analysis can successfully produce coupled tropical Pacific Sea Surface temperature-East African rainfall teleconnection from observed datasets. We further simulated East African vegetation response under future climate change as it is projected by climate models and under future climate change combined with a predicted increased ENSO intensity. Our EOT analysis highlight that climate simulations are still not good at capturing rainfall variability due to ENSO, and as we show here the future vegetation would be different from what is simulated under these climate model outputs lacking accurate ENSO contribution. We simulated considerable differences in East African vegetation growth under the influence of an intensified ENSO regime which will bring further environmental stress to a region with a reduced capacity to adapt effects of global climate change and food security.

# 1 Introduction

The 2010-2011 drought in the Horn of Africa, by some measures the worst drought in 60 years (Nicholson, 2014), is a reminder that rainfall in this politically and socioeconomically vulnerable region can fluctuate dramatically. El Niño Southern Oscillation (ENSO) influence has long been at the center of attention as a driver of this interannual fluctuations in East African rainfall (Indeje et al., 2000; Anyah and Semazzi, 2007; Nicholson, 2015), however, it is still an on-going endeavour to qualify and quantify the future behaviour of ENSO regimes under the predicted future warming (Vecchi and Wittenberg, 2010; Miralles et al., 2014). In this study we aim to identify and quantify the ENSO contribution to the East African rainfall variability in order to increase our understanding on the future response of East African vegetation to rainfall variability related to changing ENSO regimes and climate which can have dire consequences in this region in terms of food and social security.

## 1.1 East African climate

Rainfall in East African climate is primarily controlled by the seasonal passage of the Intertropical Convergence Zone (ITCZ) (Nicholson, 2000). While mean annual precipitation varies from <200 to >2000mm/year (Nicholson, 2000) and dry season length can vary from 0 to >8 months. Interannual variations in the seasonal migration of the East African ITCZ are driven to large extent by the ENSO (Ropelewski and Halpert, 1996) and its related impact through western Indian Ocean sea surface temperature (SST) anomalies (Goddard and Graham, 1999). The effect of ENSO on East African precipitation is diversified. Surface ocean warming in the western Indian Ocean (El Niño) leads to intensification and shifts of the ITCZ, bringing more precipitation to East Africa (Wolff et al., 2011), even through the direct teleconnection through the atmosphere tends to reduce rainfall (La Niña). These regions receive above average rainfall in El Niño years and below average in La Niña years during the OND months (Endris et al., 2013).

## 1.2 East African vegetation

The control ENSO exerts on East African precipitation also manifests itself on the vegetation which is contingent upon the seasonal rainfall. East Africa hosts a variety of biomes ranging from tropical rainforest to desert, however the region is mainly dominated by arid or semi-arid vegetation (Bobe, 2006). The arid and semi-arid vegetation consist of species that can tolerate aridity for several months as a result of the exceedingly seasonal precipitation (Bobe, 2006). Agricultural activities also depends on this strong seasonality as it determines the cropping times (Shisanya et al., 2011). Maize, beans, coffee, tea and wheat are among the important agricultural products of East Africa together with fruit products, and grasses for livestock (FAOSTAT).

An adaptive management of the limited resources will shape the future severity of climate change impacts on food productivity in this rainfall-reliant setup (Thornton et al., 2014). Therefore, a temporally and spatially extensive understanding of how the ecosystem dynamics in the region will respond to changing climate, and of particular concern to East Africa, to the ENSO regimes is needed. Several studies related the variability in African vegetation to ENSO events (Shisanya et al., 2011; Ivory et al., 2013; Abdi et al., 2016; Detsch et al., 2016). However, the forthcoming of this relationship has been less of a focus, partly due to our imperfect knowledge on the nature of the future ENSO response to changing climate.

### 1.3 ENSO impact on East African vegetation

An opportunity to examine the ENSO-East African vegetation relationship is by means of using predictive tools such as vegetation models which have been successfully applied to determine and forecast regional vegetation dynamics (Moncrieff et al., 2014; Scheiter and Savadogo, 2016) as well as agricultural yields (Waha et al., 2013; Dietrich et al., 2014). In this study, we used the latest climate projections from the Intergovernmental Panel on Climate Change (IPCC) $5^{th}$ assessment report for Representative Concentration Pathway (RCP) 8.5 scenario, downscaled by the Coordinated Downscaling Experiment (CORDEX) (Nikulin et al., 2012; Endris et al., 2013) to drive such a process-based dynamic vegetation model, LPJ-GUESS (Lund-Potsdam-Jena general Ecosystem Simulator). To be able to differentiate the vegetation variability due to ENSO, we removed the ENSO signal from the climate data and simulated the vegetation under the historical climate without components related to ENSO teleconnections. In the following sections, we look at the ENSO influence on East African vegetation i) under present conditions, ii) under projected future climate, and iii) under a potentially increased ENSO intensity combined with future climate change. Finally, we discuss the effects of ENSO-related vegetation variability on the carbon and hydrological cycles, and its significance for mitigation efforts in the region.

## 2 Methods

### 2.1 The LPJ-GUESS model

We used the dynamic vegetation model LPJ-GUESS (Lund-Potsdam Jena General Ecosystem Simulator) for our study. LPJ-GUESS is a mechanistic model in which ecosystem processes are simulated via explicit equations and is optimised for regional to global applications (Smith et al., 2001; Sitch et al., 2003; Gerten et al., 2004). Vegetation dynamics are simulated as the emergent outcome of growth, reproduction, mortality and competition for resources among woody plant individuals and herbaceous vegetation.

The simulation units in this study are plant functional types (PFTs) distinguished by their growth form, phenology, photosynthetic pathway ($C_3$ or $C_4$), bioclimatic limits for establishment and survival and, for woody PFTs, allometry and life history strategy. The simulations of this study were carried out in 'cohort mode', in which, for woody PFTs, cohorts of individuals recruited in the same patch in a given year are represented by a single average individual, and are thus assumed to retain the same size and form as they grow. A sample instruction file used to run LPJ-GUESS in this study with all the parameters listed can be found under github.com/istfer/ENSOpaper/ins.

Primary production and plant growth follow the approach of LPJ-DGVM (Sitch et al., 2003). Population dynamics (recruitment and mortality) are influenced by available resources and environmental conditions, and depends on demography and the life history characteristics of each PFT (Hickler et al., 2004). Disturbances such as wildfires are simulated based on temperature, fuel load and moisture availability (Thonicke et al., 2001). Litter arising from phenological turnover, mortality and disturbances enters the soil decomposition cycle. Decomposition rates depend on soil temperature and moisture (Sitch et al., 2003). Soil hydrology follows Gerten et al. (2004). A more detailed description of LPJ-GUESS is available in Smith et al.

(2001). We used LPJ-GUESS version 2.1 which includes the PFT set and modifications described in Ahlström et al. (2012). LPJ-GUESS has already been successfully applied and validated to match present-day and mid-Holocene biome distributions of East Africa as suggested by data for both periods (Fer et al., 2016).

## 2.2 Datasets Tracking ENSO and regional vegetation

To isolate the ENSO signal contribution to East African precipitation, we conducted an Empirical Orthogonal Teleconnections (EOT) analysis between sea surface temperatures (SSTs) in the tropical pacific ocean and precipitation over East Africa (see section Identifying the ENSO signal). For historical extraction (1951-2005), we use monthly National Oceanic and Atmospheric Administration Extended Reconstructed Sea Surface Temperature (NOAA ERSST) V4 dataset (Huang et al., 2014; Liu et al., 2014), available on $2°$x$2°$ global grids as a predictor field. As the response series, we used monthly Climatic Research Unit Time Series (CRU TS) 3.20 dataset (Harris et al., 2014), available on $0.5°$x$0.5°$ global grids.

### 2.2.1 LPJ-GUESS datasets

LPJ-GUESS requires monthly climate (temperature, precipitation, cloud cover), atmospheric $CO_2$ concentration and soil texture as input data. For historical period (1951-2005), we used monthly CRU TS 3.20 climate data. We chose these years for all historical analysis throughout the study as the historical simulations of CORDEX outputs are available for this period. For future projections (2006-2100), we used the outputs from the Coordinated Regional Climate Downscaling Experiment (CORDEX) program for the Africa domain. For reporting historical (1951-2005) and future (2006-2100) period, we adhered to the CORDEX division of years for interpretability and reproducibilityreasons. For the future scenario, we chose the baseline high emissions Representative Concentration Pathways (RCP) 8.5 scenario under the assumption that climate mitigation targets will not be met (Moss et al., 2010; Riahi et al., 2011). CORDEX, downscaled global climate models (GCMs) by using regional models, and the outputs are available from ESGF-CoG data portal (https://esgf-node.llnl.gov/search/esgf-llnl/). For East African climate, we took the ensemble mean of 9 models for future projections of RCP 8.5 scenario as these are the available, dynamically downscaled climate model outputs by the CORDEX project: CCCma CanESM2, CERFACS CNRM-CM5, QCCCE CSIRO Mk3-6-0, ICHEC EC-EARTH, IPSL CM5A-MR, MIROC5, MPI ESM-LR, NCC NorESM1-M, NOAA GFDL-ESM2M (Full names of the models are given in the Appendix). Instead of working with individual models we decided to drive our simulations with ensemble means as it has been shown to outperform individual models and show a better agreement with data (Endris et al., 2013). RCP 8.5 compatible atmospheric $CO_2$ values were also used as provided by NOAA-GISS experiment (Nazarenko et al., 2015).

### 2.2.2 Bias correction

To eliminate biases originating from using CRU climate dataset for present and model simulations for future, we subtract the 1951-2005 climatology of downscaled GCM ensemble from the 1951-2100 time series of the ensemble and add the anomalies on CRU 1951-2005 climatology. This way we will able to have a meaningful comparison between CRU-driven and GCM-

driven vegetation model outputs while keeping the climate variability from the GCM simulations. We should note here, that
this would not change the ENSO signal we will retrieve from the GCM outputs (see next section) because we de-season and
work with anomalies of the data field for our EOT analysis.

### 2.2.3 Future Pacific SSTs

For future pacific SSTs, we used outputs from GCM simulations of the same models listed above for RCP 8.5, except ICHEC
EC-EARTH which was not available from the data portal at the time. However, these GCM outputs were not downscaled and
standardized in terms of spatial resolution (they were all available in monthly time steps in terms of temporal resolution). We
created raster files from these outputs and using the 'raster-package' (R, 2015), we resampled these rasters to brought them to
the same spatial resolution as NOAA ERSST V4 dataset, and we took the ensemble mean.

### 2.3 Identifying the ENSO signal

Here we first identify the ENSO signal as a driver for monthly East African precipitation variability over the historical period
(1951-2005). To do this we investigate the teleconnectivity between the SSTs in the tropical Pacific Ocean and precipitation
over East Africa by using empirical orthogonal teleconnections (EOT). The method is explained by van den Dool et al. (2000)
in detail, and Appelhans et al. (2015) implemented the original algorithm in R, 'remote' package. Here, we only briefly present
the major steps of the EOT analysis:

### 2.3.1 Empirical Orthogonal Teleconnections (EOT)

In the EOT analysis, we aim to establish an explanatory relationship between the temporal dynamics of a (predictor) domain
and temporal variability of another (response) domain. Such predictor and response domains consist of gridded time series
profiles: in this study the gridded monthly SST time series of the tropical pacific as predictor and gridded precipitation time
series of East Africa as the response. Then, the first step of EOT analysis is to regress these time series of each predictor domain
grid ($N_p$) against the time series of each response domain grid ($N_r$) (Appelhans et al., 2015). This will result in a ($N_p$ x $N_r$)
number of regression fits after which we can calculate the sum of coefficients of determination per predictor grid (ending up
with $N_p$ sum of coefficients of determination values). Then, the grid with the highest sum will be identified as the 'base point'
of the leading mode as it explains the highest portion of the variance in the response domain (Appelhans et al., 2015). The
time-series at this base point is referred as the leading teleconnection, or hereafter as the first EOT.

### 2.3.2 Screening for ENSO signal

We applied the EOT method on de-seasoned and de-noised data fields in order to retrieve a low frequency signal such as
ENSO: here we used the SSTs in the tropical pacific ocean as predictor and precipitation over East Africa as response. Then
we proceeded to calculate the SSTs modes that are most affecting for East African rainfall variability. We found the $1^{st}$ EOT

to be the ENSO signal. We compared this EOT with Nino3.4 index to see whether we were able to isolate the ENSO signal. The commented code used for all methods is publicly available on Github (github.com/istfer/ENSOpaper).

Before moving on to identifying future pacific sea surface temperature-East African precipitation interactions, we applied the same extraction to historical GCM outputs (simulations) to see whether we can identify a similar relationship from GCM products. Finally, we prepared the model drivers with the modified ENSO signal we identified from the future simulations (see next section) and ran the model with these datasets (here we focused on precipitation data only, while precipitation varies in these simulations and the others -temperature- were kept as they were in the climate datasets: present - CRU TS 3.2, future - CORDEX ensemble de-biased using CRU as explained above).

## 2.4 Removing and intensifying the ENSO signal

In order to investigate the contribution of the ENSO signal to East African precipitation, we removed the ENSO signal and explored the rainfall pattern with and without ENSO contribution as well as the resulting vegetation changes calculated by LPJ-GUESS. We used the 'remote' package which specifically implements the EOT analysis and keeps track of calculated values in a structured workflow: The rainfall we are left with after removing the first EOT mode (which we identified as the ENSO signal) becomes the rainfall behaviour without ENSO contribution (within the 'remote' package, this calculation of the residuals is automatically available after the calculation of the EOT modes). Therefore, if we take the difference between these residuals and the initial de-seasoned and de-noised data, this will give us the amount that we need to subtract from the raw data field to obtain the rainfall behaviour without ENSO contribution. The steps are explained below as pseudocode:

i) Deseason and denoise the response and predictor fields.

$EA_{r,ds,dns}$: East African precipitation (response domain). Subscripts indicate raw, deseasoned, deseasoned and denoised respectively.

$PAC_{r,ds,dns}$: Tropical Pacific Ocean Sea Surface Temperatures (SSTs) (predictor domain). Subscripts indicate raw, deseasoned, deseasoned and denoised respectively.

$$EA_{ds} = deseason(EA_r) \qquad\qquad PAC_{ds} = deseason(PAC_r) \tag{1}$$

$$EA_{dns} = denoise(EA_{ds}) \qquad\qquad PAC_{dns} = denoise(PAC_{ds}) \tag{2}$$

ii) Conduct Empirical Orthogonal Teleconnection (EOT) analysis:

$$EOT_{modes} \longleftarrow EOT(EA_{dns} \sim PAC_{dns}) \tag{3}$$

Here the $EOT_{modes}$ object can be thought as a list that stores both the time series of the modes, the reduced fields obtained after the removal of each mode, slopes and intercepts of the fields (for more details see Appelhans et al. (2015).

iii) Calculate the difference ($Diff$) between the de-seasoned, de-noised data ($EA_{dns}$) and the rainfall behaviour without ENSO contribution from the information that is already stored in the resulting $EOT_{modes}$ object (ENSO signal is the first

mode, therefore the rainfall behaviour we are left without ENSO will be the $EA_{modes,rr1}$ where subscript $rr1$ indicating 'response residual' after the removal of the first EOT mode:

$$Diff = EA_{dns} - EA_{modes,rr1} \tag{4}$$

iv) If we subtract this difference from the initial raw response field ($EA_r$), we will obtain the East African precipitation without ENSO contribution ($EA_{r,woENSO}$):

$$EA_{r,woENSO} = EA_r - Diff \tag{5}$$

v) As EOT analysis is basically a regression analysis, we can also obtain the ENSO contribution ($Diff$) from the regression equation as shown below (which will become handy when we insert back the intensified ENSO signal):

$$Diff = EOT_{modes,eot1} * EOT_{modes,ri1} - EOT_{modes,rs1} \tag{6}$$

Here $EOT_{modes,eot1}$, $EOT_{modes,ri1}$ and $EOT_{modes,rs1}$ refer to the EOT time-series of the $1^{st}$ mode (the ENSO signal),

intercept of and slope of the response field calculated for the $1^{st}$ mode (Appelhans et al., 2015).

vii) Then, it is possible to modify the future ENSO signal ($EOT_{modes,eotF}$) obtained from EOT analysis on simulation datasets, re-calculate its contribution to the East African rainfall ($Diff_{new}$) and add this amount back on the precipitation data without ENSO signal ($EA_{r,woENSO}$) to obtain new precipitation amounts ($EA_{r,new}$) due to new signal. We can later use this $EA_{r,new}$ as the future precipitation input to our vegetation model to drive future simulations.

$$Diff_{new} = EOT_{modes,eotF} * EOT_{modes,ri1} - EOT_{modes,rs1} \tag{7}$$

$$EA_{r,new} = EA_{r,woENSO} + Diff_{new} \tag{8}$$

Here it is noticeable that slope(s) and intercept(s) would also have been different if the ENSO signal was changed ($EOT_{modes,eotF}$). However, this simplification is adequate for experiments in this paper. Moreover, we used the intercept and slope we retrieved from the EOT analysis on observational datasets while re-calculating the new difference ($Diff_{new}$) due to intensified ENSO

signal. Because the East African rainfall patterns explained by Tropical Pacific SSTs in the GCM simulations are different from observations (Figure A1 and A2). By using slopes and intercepts obtained from the observational data we were also able to preserve the more accurate patterns in rainfall differences.

viii) Finally, we obtained the modified ENSO signal ($EOT_{modes,eotF}$) in Eq. (7) by detrending (fitting a LOWESS smoother and removing it from the signal) and multiplying the ENSO signal we extracted from the future simulations (deseasoned and denoised GCM simulations for East African rainfall, $EA_{dns,ftr}$, and Tropical Pacific SSTs, $PAC_{dns,ftr}$) with a coefficient ($k = 3$) such that the peaks of the new signal would be as strong as the observed anomalies ($\pm$ 2.5 °C, Figure 1 and S3). For the code of this step see IdentifyModifyFutureENSO.R script at github.com/istfer/ENSOpaper.

$$EOT_{modes,ftr} \longleftarrow EOT(EA_{dns,ftr} \sim PAC_{dns,ftr}) \tag{9}$$

$$EOT_{modes,eotF} = k * detrend(EOT_{modes,ftr,eot1}) \tag{10}$$

## 2.5 Results

## 2.6 EOT analysis - extracting the ENSO signal

We compared the first EOT mode extracted after de-seasoning and de-noising the fields as explained by Appelhans et al. (2015) to the Nino-3.4 index recorded (Figure 1). The high correlation between the two (R = 0.90) confirms that we were able to extract the ENSO signal by conducting the EOT analysis. On the predictor domain (Tropical Pacific SSTs), the Nino-3.4 region found to be the area which explains the most variance in the response domain (East African precipitation) as expected (Figure A1). The time series of the first EOT explains 0.85% of the rainfall variability over the analyzed period here (1951-2005). This small amount is not surprising, because East African precipitation follows a strong seasonal pattern following the position of the Intertropical Convergence Zone (ITCZ) within the year. Therefore, seasonality alone explains most of the variability in East African rainfall. In addition, due to the complex topographical setting of the region, local conditions play a major role in the variation of the rainfall. Still, when we de-season and de-noise the raw data fields to identify low-frequency signals such as ENSO, the ENSO signal emerged as the most important teleconnection between tropical pacific SST anomalies and East African precipitation.

Having successfully extracted the ENSO signal from the observation datasets, we applied the same procedure with the outputs of the climate models. We used an ensemble of SSTs from 8 GCM outputs as the predictor field and an ensemble of rainfall from 9 GCMs downscaled by CORDEX as the response domain. The comparison between the calculated first EOT time series to the Nino-3.4 index observed was much poorer (R=0.19) (Figure 1) which indicates that GCMs are not capturing the coupled Pacific SST - East African rainfall teleconnection. Another striking feature that can be observed in Figure 1 is the smoothness of the time series of the 1st mode calculated from the EOT analysis on ensembles of climate model outputs when compared to the recorded index and the calculated ENSO signal from the observation datasets. In other words, the ENSO signal retrieved from the EOT analysis on the climate model outputs is nowhere near strong as the others. According to this signal obtained from the simulation datasets, the only ENSO events that happened during 1951-2005 period were in the 'weak' category (Figure 1). Finally, the calculated patterns were different than the EOT analysis on observed datasets (the

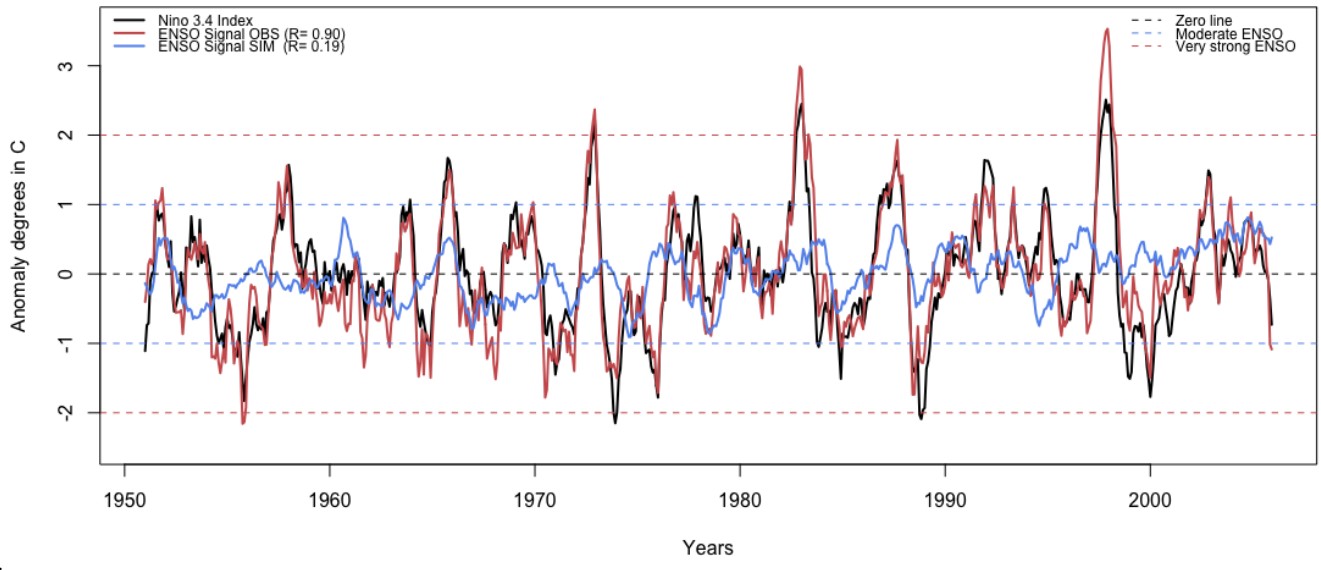

Nino 3.4 vs EOT modes

ii

**Figure 1.** The comparison between Nino 3.4 Index recorded by NOAA (black line), time series of the 1st mode obtained from the EOT analysis on observed CRU-NOAA datasets (red), and time series of the 1st mode obtained from the EOT analysis on ensembles of the climate model simulations (blue). Black dashed line: Zero line. Blue dashed lines $\pm 1.0°$ anomaly thresholds for categorizing moderate ENSO events. Red dashed lines: $\pm 2.0°$ anomaly thresholds for categorizing very strong ENSO events.

corresponding figure is given in the Appendix, Figure A2): The areas where the sum of the coefficients of determination were the highest were again situated around Nino 3.4 region but closer to the Nino 4 region this time (Figure A2 - left panel). Spatially, the north-eastern and central parts of the response domain are the most explained whereas previously it was more centralized around the coastal equatorial parts of the region (Figure A2 - right panel).

## 2.7 Historical simulations with and without the ENSO signal

After calculating the ENSO signal, we removed the amount due to ENSO from the East African precipitation (CRU precipitation), and simulated East African vegetation using both datasets ($CRU_{normal}$ and $CRU_{withoutENSO}$) to see its effect on vegetation. As it can clearly be seen from Figure 1, impact of ENSO signal is not the same everywhere on the East African domain, which means removing ENSO signal would have differential effects on the rainfall amount. Regional maps of rainfall anomalies for the strongest three El Niño (1972, 1982, 1999) and La Niña (1973, 1975, 1988) events between 1951-2005 period are given in Figure 2. Here we show what the rainfall would be if there were not any influence by the Pacific SSTs particularly during these three years. Especially the coastal Kenya and Tanzania experience a strong change in the amount of rainfall they receive: During the El Niño periods, these parts of East Africa receive up to 200 mm yr$^{-1}$ more rain other than they would

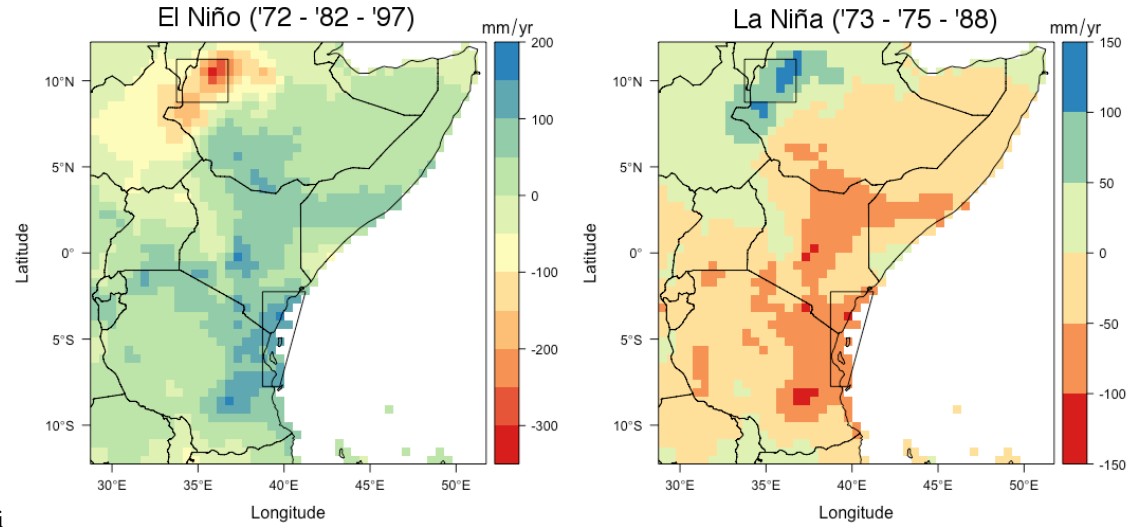

ii

**Figure 2.** Regional maps of anomalies ($\mathrm{mm\ yr^{-1}}$) for the strongest three (1972, 1982, 1997) El Niño (left) and (1973, 1975, 1988) La Niña (right) events between 1951-2005 period (anomalies were calculated by subtracting precipitation without ENSO contribution from precipitation with ENSO contribution). Northern inner and southern coastal transects chosen for reporting results on vegetation simulations.

receive, while they receive $\approx 100\ \mathrm{mm\ yr^{-1}}$ less rain during the La Niña years. The impact is the opposite for western part of

Ethiopia : receiving $\approx 200\ \mathrm{mm\ yr^{-1}}$ less rainfall during El Niño years, while $\approx 100\ \mathrm{mm\ yr^{-1}}$ more during La Niña years. To provide a closer look to the impacts of ENSO related variability on vegetation, we report the results on vegetation simulations within the two transects where we see the strongest impacts over these two oppositely behaving, coastal and northwestern, regions (Fig. 2).

We drove the dynamic vegetation model once with CRU dataset as is and once with CRU dataset with removed ENSO

contribution. Results are reported for the previously mentioned north and south sites in Figure 3 and Table 1. Outputs from the northern-inner part show more variability within the chosen grid-cells for this region. Indeed, this region is on the western edge of Ethiopian Plateau, with a transition of biomes from mountainous forests to woodlands and savannas (Fer et al., 2016). As the rainfall patterns in relation to ENSO signal was the opposite between these regions (Figure 2), we expect to see that the response of these regions to the removal of the ENSO signal to be opposite, and this is indeed what we see in Figure

3: While outputs such as net primary productivity (NPP), net ecosystem exchange (NEE), and surface runoff (RUNOFF) for northern site were less than otherwise they would be for El Niño events, they would be higher La Niña events. And the opposite behaviour is true for the southern site.

In order to test whether the difference between the vegetation simulated under climate with ENSO contribution, and the vegetation simulated under climate with removed ENSO contribution, we conducted a paired t-test on the outputs. The results

(Table 1) show that except NEE for northern sites, all differences between the vegetation simulated with and without ENSO

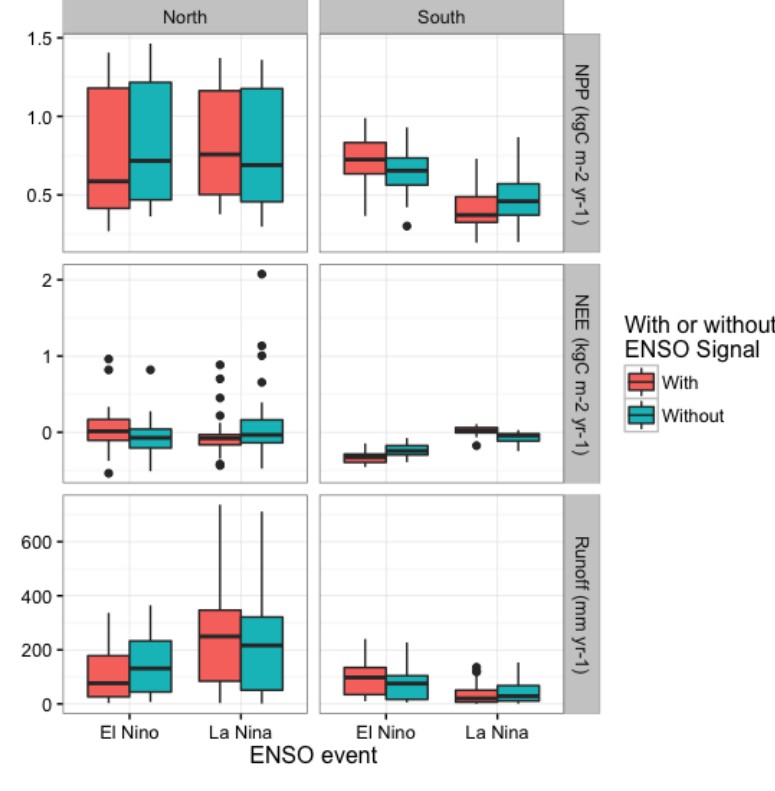

ii

**Figure 3.** Carbon and water fluxes from north and south transects, simulated under climate with and without ENSO contribution, for the strongest three (1972, 1982, 1997) El Niño and (1973, 1975, 1988) La Niña events between 1951-2005 period. Top panel: Net Primary Productivity (NPP). Middle: Net Ecosystem Exchange (NEE). Bottom: Total runoff. Locations of the northern-inner and southern-coastal sites are given in Figure 2.

impact were significant. In summary, ENSO contribution is significantly affecting the East African vegetation and we would expect different vegetation if there were no ENSO events.

## 2.8 Future simulations with and without the intensified ENSO signal

We conducted the same paired t-test for the north and south sites for the future simulations (Table 1). In the northern site where intensified signal leads to less (more) NPP during El Niño (La Niña) years, the mean difference is -52 (+32.6) gC m$^{-2}$ yr$^{-1}$ between the vegetation simulated under future climate with and without intensified ENSO signal. In the southern site where intensified signal leads to more (less) NPP during El Niño (La Niña) years, the mean difference is +49.1 (-101.1) gC m$^{-2}$ yr$^{-1}$ between the vegetation simulated under future climate with and without intensified ENSO signal. While the mean differences for NEE were not significant at the northern site, southern site stores 112.7 (173.1) gC m$^{-2}$ yr$^{-1}$ more (less) carbon under the intensified ENSO scenario during the El Niño (La Niña) years.

**Table 1.** Paired t-test results to test whether there is a significant difference in the vegetation simulations that are driven with and without ENSO contributions for the three strongest ENSO events during the historical period (1951-2005) and with and without intensified ENSO signal for the strongest ENSO events during the future period (2006-2100). Italics indicate insignificant differences according to p=0.05 threshold. Significant p-values indicate rejection of the $H_0$ in favor of the alternative, that is true difference in means is not equal to 0.

| | | NPP $(\text{kgC m}^{-2}\text{ yr}^{-1})$ | | NEE $(\text{kgC m}^{-2}\text{ yr}^{-1})$ | | RUNOFF $(\text{mm yr}^{-1})$ | |
|---|---|---|---|---|---|---|---|
| | | El Niño | La Niña | El Niño | La Niña | El Niño | La Niña |
| Historical | N | $p < 0.05$ md: -0.056 | $p < 0.05$ md: 0.035 | *$p = 0.089$* | *$p = 0.1$* | $p < 0.05$ md: -41.35 | $p < 0.05$ md: 22.21 |
| Historical | S | $p < 0.05$ md: 0.084 | $p < 0.05$ md: -0.074 | $p < 0.05$ md: -0.088 | $p < 0.05$ md: 0.087 | $p < 0.05$ md: 19.41 | $p < 0.05$ md: -10.74 |
| Future | N | $p < 0.05$ md: -0.052 | $p < 0.05$ md: 0.033 | *$p = 0.93$* | *$p = 0.58$* | $p < 0.05$ md: -10.91 | $p < 0.05$ md: 46.97 |
| Future | S | $p < 0.05$ md: 0.049 | $p < 0.05$ md: -0.101 | $p < 0.05$ md: -0.113 | $p < 0.05$ md: 0.173 | $p < 0.05$ md: 5.66 | *$p = 0.06$* |

Location of North (N) and South (S) sites are shown on Figure 2. p: p-value, md: mean of the differences. NPP: Net Primary Productivity, NEE: Net Ecosystem Exchange, RUNOFF: Surface runoff

Another noteworthy output is that, the northern site has a lot more runoff during the La Niña years under the intensified ENSO scenario. This is especially clear on Figure 4 where spatial patterns of the differences in the simulated future vegetation under RCP 8.5 scenario with and without intensified ENSO are shown. The opposite behaviour of the northern parts of East Africa under El Niño vs. La Niña conditions can also be observed on NPP and RUNOFF figures, wheres for NEE differences a particular pattern is not emergent. This is mainly because NEE values can themselves be negative (flux to ecosystem) and positive (release to atmosphere).

The opposite temporal behaviours of the northern and southern transects are also clear in Figure 5 which shows the time series of the differences between simulated NPP, NEE and RUNOFF under climate drivers with and without intensified ENSO signal. In line with the characterized behaviours above, we simulated higher (lower) NPP for the southern transect (red line) for the El Niño (La Niña) years under the intensified scenario, whereas the opposite is true for the northern transect (black line). The higher amplitude of RUNOFF difference for the northern transect is notable in the bottom panel (Figure 5).

## 3 Discussion

### 3.1 Identifying and intensifying the ENSO Signal

East African rainfall variability and especially contribution of the ENSO was investigated before citepIndeje2000, SchreckSemazzi2004. Here we used a different method, Empirical Orthogonal Teleconnections (EOT) analysis to quantitatively calculate

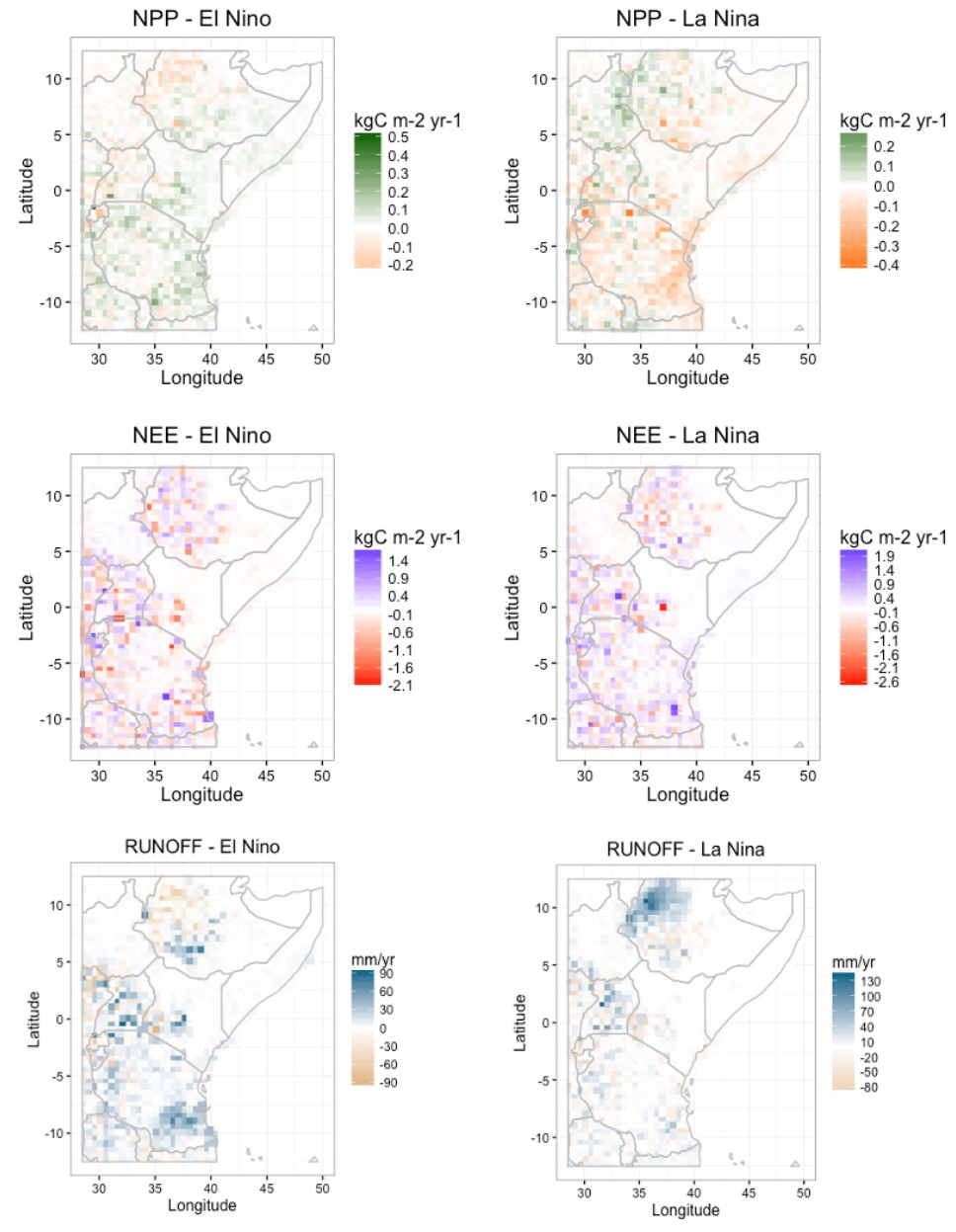

ii

**Figure 4.** Simulated future differences in the NPP, NEE and RUNOFF between with and without intensified ENSO runs. (Left) Mean differences for the strong El Niño years ($\geq$ +1.5 °C) (2025, 2026, 2077) were calculated by subtracting the GCM-ensemble driven simulations without modification from the GCM-ensemble driven future simulations with intensified ENSO signal. (Right) Same for strong future La Niña events ($\leq$ -1.5 °C) (2039, 2049, 2084).

the ENSO contribution and found the spatial correlation patterns over the East Africa region to be in agreement with previ-

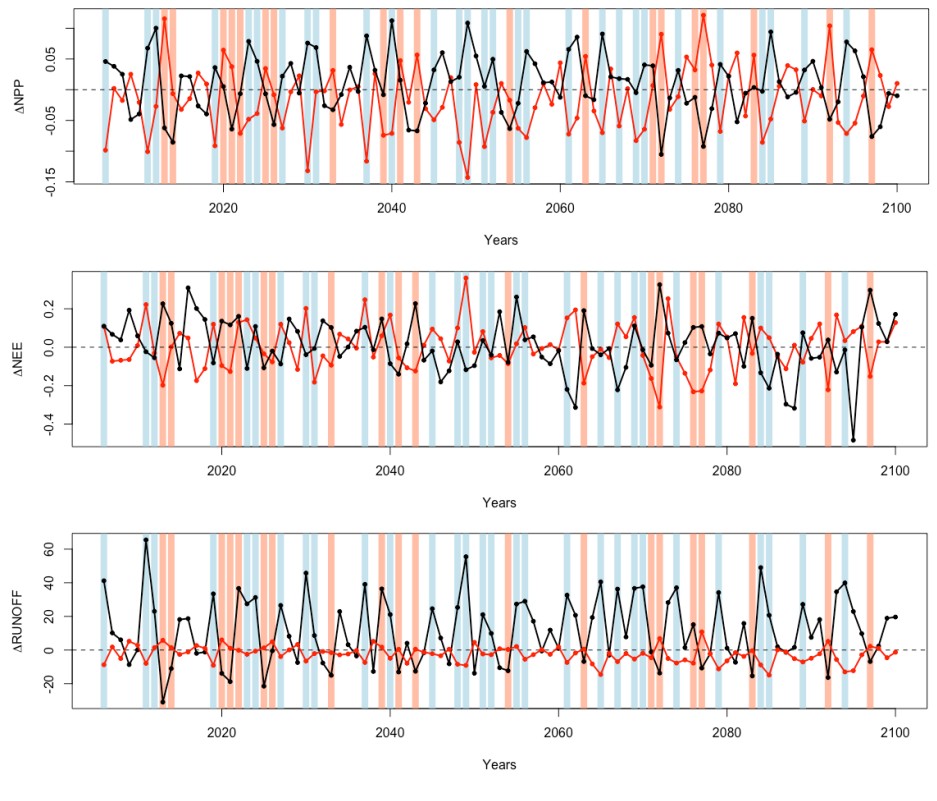

ii

**Figure 5.** Temporal differences in the NPP, NEE and RUNOFF according to future simulations with and without intensified ENSO contribution ($\Delta$ = With Intensification - Without Intensification). Black line: Northern transect, Red line: southern transect. Vertical blue lines: All moderate (< -1.0 °C) La Niña years identified for the future period (2006-2100), Vertical pink lines: Moderate (> 1.0 °C) El Niño years. The units are same as Figure 4.

ous studies who independently looked at Pacific SST drivers for East African precipitation (Anyah and Semazzi, 2007). The ENSO signal identified through this method was also showing strong correlation with NOAA Nino3.4 index, which means

EOT method was a suitable choice for our analysis.

Using the EOT method, we presented a relatively conservative estimate of ENSO variability in East African rainfall, because we considered the direct Tropical Pacific teleconnection only. However, there are accompanying changes: ENSO events are linked to Indian Ocean Dipole, which more directly influences EA rainfall (Black et al., 2003). It has been suggested that subsequent to ENSO triggering, internal Indian Ocean dynamics could take over. More specifically, East African rainfall

increases as the western Indian Ocean gets warmer which is often associated with ENSO forcing. However, warmer western Indian Ocean can weaken the rains when it interacts with southeasterly atmospheric circulations (Schreck and Semazzi, 2004). The exact relationship and discrepancies between IOD and ENSO behaviours are yet to be revealed (Lim et al., 2017). Still, we found that the ENSO-East Africa connection to be robust as previous studies (Indeje et al., 2000; Anyah and Semazzi, 2007)

and did not delve into IOD relationship. Also, we were motivated by the previous studies that have identified ENSO influence

to be important in dryland vegetation dynamics (Ahlström et al., 2015; Abdi et al., 2016). Hence, we focused on reporting more comparable results with those. Another factor that could affect our estimations is atmospheric latency. In our analysis, we did not consider any time lags for the tropical pacific SST anomalies and East African precipitation teleconnection, but a time lag can be expected due to atmospheric circulation processes, and the influence of SST anomalies might not develop instantaneously. Therefore if we account for this time lag, we might explain even more of the rainfall variance. For a more

comprehensive study of SST influences on East African rainfall see Appelhans and Nauss (2016).

The EOT method, which is shown here to be effective on the historical observations, produced different East African rainfall variability patterns due to Pacific SSTs when GCM outputs were used. Also the ENSO signal retrieved was much weaker than the one extracted from the observation datasets in terms of both ENSO event strength and the match (correlation) with the Nino 3.4 index. As a preliminary investigation (not shown), we conducted the EOT analysis across mixture of

observed-simulated datasets: $Pacific\ SSTs_{observed}$ (NOAA ERSST) - $East\ African\ precipitation_{simulated}$ (CORDEX), and $Pacific\ SSTs_{simulated}$ (GCMs) - $East\ African\ precipitation_{observed}$ (CRU). The ENSO signal retrieved from the $Pacific\ SSTs_{observed}$ - $East\ African\ precipitation_{simulated}$ pair was a better match with Nino 3.4 index than the one extracted from the simulated-simulated pair but still worse than the one extracted from observed-observed dataset pair, whereas ENSO signal retrieved from the $Pacific\ SSTs_{simulated}$ - $East\ African\ precipitation_{obsrved}$ pair was not a better match to Nino 3.4 index than the one extracted from the simulated-simulated pair. This quick test indicated that the GCM simulated Tropical Pacific SSTs are the main source of the poor teleconnection identified from the simulated-simulated pair and a dynamic downscaling of the tropical Pacific SSTs might improve the ocean-atmosphere coupled teleconnection. However, more

formal tests are needed to conclude on this matter, which was beyond the scope of this study.

## 3.2 Present-day simulations

Despite the fact that our estimation of ENSO contribution to the East African interannual rainfall variability was conservative, the precipitation difference between with and without ENSO contribution was equivalent to one or even two rainy months for some of the grid cells. These regions already receive a small amount of rainfall and even minor differences are critical for

agricultural food production and the productivity of the natural ecosystem that sustains a large biodiversity. We found up to 0.1 kgC m$^{-2}$ yr$^{-1}$ mean difference in NPP in the southern parts of the region solely due to ENSO contribution.

We found that ENSO influence on net ecosystem exchange is also prominent in the semi-arid ecosystems of East Africa. Especially, in southern-coastal parts, ecosystem releases more to the atmosphere during La Niña events whereas it would store more carbon otherwise. This would also have implications on global carbon cycle as it has previously been found that

regional response of semi-arid ecosystems, mainly occupying low latitudes, play an important role in determining the trend in $CO_2$ uptake by terrestrial ecosystems (Ahlström et al., 2015). For instance, La Niña events are associated with large carbon sinks in Australian semi-arid ecosystems due to increased precipitation and 2011 anomaly in global carbon sink was mainly attributed to the response of Australian ecosystems (Poulter et al., 2014). While semi-arid ecosystems of East Africa might play a smaller role than Australian ones (simply due to the difference in the area they cover), it would still influence the magnitude

and trend of the global carbon sink by terrestrial ecosystems. Furthermore, Forzieri et al. (2017) report the importance of the interplay between vegetation cover (in terms of Leaf Area Index, LAI) and surface biophysics, finding an amplification of their relationship under extreme warm-dry and cold-wet years. Here we found that the ENSO contribution impacts the temporal LAI variability in East Africa considerably (Figure A5), presenting a good example of such temporal variations that can play significant roles in modulating key vegetation-climate interactions. According to the analysis by Forzieri et al. (2017), the magnitudes of differences we found in our study due to accounting for an intensified ENSO signal are influential on the surface energy balance components such as longwave outgoing radiation, latent heat flux and sensible heat flux. Our findings reiterate the importance of considering ENSO contribution in carbon and energy budget calculations for any region that is influenced by ENSO variability.

Here we also report ENSO influence on surface runoff as excess runoff response causes problems in East Africa. In this region, Rift Valley Fever (RVF) and Malaria outbreaks are threatening the livelihood of the society and these vector-borne diseases are transmitted by mosquitoes who breed in flooded low-lying habitats (Meegan and Bailey, 1989; Kovats et al., 2003; Hope and Thomson, 2008). For example, a major RVF outbreak during late 1997 to early 1998 has been linked to the heavy and prolonged rains that are associated with 1997-98 El Niño event (Trenberth, 1998), in agreement with our results where we found that the southern coastal site experiences higher runoff during El Niño events than otherwise it would do.

Another important ecological factor to be considered for East African vegetation dynamics is fire. The fire occurrence in LPJ-GUESS depends on the atmospheric temperature values, and moisture and litter availability. Therefore, although we did not calibrate LPJ-GUESS fire parameters for East Africa or explicitly changed fire regimes under any of the scenarios, the model simulated the changes in fire behaviour due to different environmental states implicitly. More specifically, for the southern coastal part, a higher mean expected return time of fire was simulated during the El Niño years for simulations with ENSO contribution than without due to higher moisture availability during ENSO years for this region (not shown). For the same site, the opposite was true for La Niña years, and the whole behaviour was reversed for the northern site. A more sophisticated fire-ENSO-vegetation interplay can be further investigated using models that have an individual level representation of fire response such as aDGVM2 (Scheiter et al., 2013).

In this study, we did not further calibrate the LPJ-GUESS PFT parameters as it has been calibrated and validated for the region by previous studies (Doherty et al., 2010; Fer et al., 2016). It is possible that these point estimate values do not capture the uncertainties associated with the PFT parameters. However, previous studies have shown LPJ-GUESS parameters to be robust (Zaehle et al., 2005; Doherty et al., 2010). Besides, as we used the same set of parameters for all runs, the discrepancies simulated with and without ENSO contribution would still hold. As LPJ-GUESS spins up from bare ground, we also do not expect much uncertainty influencing the model predictions with and without ENSO contribtuion due to initial conditions. On the other hand, we expect the driver uncertainty to dominate the uncertainty around model predictions. However, that is exactly what we aimed at to quantify in this study as being discussed in the following sections.

### 3.3 Scenario selection and future simulations

In the results for the future simulations, the total surface runoff and NPP responses were considerably underestimated. Under the intensified ENSO scenario, an excessive amount of runoff is simulated for the northern parts during La Niña years and for the southern parts during El Niño years, which would exacerbate the disease events in the region. Likewise, the simulated low amounts of runoff for the northern parts during El Niño years indicate drought events in this parts of the region. This effect can also be seen in the simulated NPP responses which reduces considerably for the northern parts during El Niño years. Furthermore, the amounts we calculated here agree well with previous studies showing changes in NPP supply associated with ENSO events in sub-Saharan African drylands (Abdi et al., 2016).

The regions identified to be impacted by ENSO the most, are also the regions that currently undergo the highest woody vegetation decrease and human population increase in East Africa according to the analysis by Brandt et al. (2017). In our future simulations, we simulated increase in woody vegetation LAI due to climate change (Figure A4) in those regions of East Africa. It requires further analysis to say whether this anthropogenic reduction in woody vegetation could be met by future climate and atmospheric $CO_2$ related increase. However, it reinforces the essentiality of accounting for ENSO influence as independent analyses show increasing stress over this region.

In this study, we chose RCP8.5 as our future warming scenario for two reasons: i) we aimed to follow the current trajectory which is pointing beyond RCP 8.5 scenario given the observed trends (Sanford et al., 2014), ii) we intended to capture the furthest range presented by RCPs as that is the extent to be considered for the assessment of ecosystem responses and mitigation efforts. However, we found that the ENSO signal as identified by the EOT method to be very weak in the GCM outputs and for the future simulations we intensified the ENSO signal such that very strong ENSO years can also be experienced as it is the real-world case. It could be argued that we did not even applied an extra intensification due to RCP8.5, and this discrepancy would hold regardless of the future scenario. Considering that we are expected to experience even stronger ENSO events in the future than today (Cai et al., 2014, 2015) we could have intensified this signal even more. However, our results with this realistic intensification already shows the importance of capturing atmosphere-ocean teleconnections in climate simulations for reliable future simulations of the ecosystems. We simulated large differences in future ecosystem responses under our 'intensified' ENSO scenario, as large as the differences we calculated for the present-day with and without ENSO simulations. In other words, if we were to predict vegetation response to future climate change by using GCM outputs as they are, it would be as if simulating the present-day vegetation with climate data without any ENSO contribution.

Apart from the temporal and strength mismatch, the GCM simulations are also producing different spatial patterns for tropical Pacific SST-East African rainfall teleconnection. Therefore, in our modification we chose to correct for this spatial pattern by using the relationships we obtained from the observed datasets as this correction did not influence the temporal behavior and the peakiness of the ENSO signal retrieved from the GCM simulations. As a result, our findings can be compared for present day patterns directly.

Another finding in our study regarding the spatial patterns was that, while the region that explains the most variability in East African rainfall is closer to the Nino-3.4 region in our historical analysis, it shifts towards the Nino-4 region in the EOT

analysis with GCM outputs. In our methodology the coupling of tropical Pacific Sea Surface temperature-East African rainfall variability emerges from the data, and this shift in the influence region agrees well with previous studies that identify an increase in the intensity of Central-Pacific (CP) ENSO in the future from GCM outputs (Kim and Yu, 2012). While CP ENSO is thought to be forced by changes in the atmospheric circulation, mechanism for Eastern-Pacific ENSO is rather associated with thermocline variations in the oceanic circulation (Yu et al., 2010), and the seasonal impacts produced by these two types of ENSO could differ. For example, wetter patterns of EP El Niño events in East Africa might not occur under CP El Niño events and, CP La Niña events could induce drier conditions in the southern parts of the region than EP La Niña events (Wiedermann et al., 2017) which could result in prolonged drought events for the East Africa region. Future work with further discrimination of CP-EP event types could help better anticipate the ecosystem responses to such seasonal extremes.

## 4    Conclusions

In this study, we translated the lack of ability of GCMs to account for ENSO teleconnections into quantified discrepancies in terms of ecosystem responses. We investigated the relationship between interannual East African rainfall variability and ENSO events using Empirical Orthogonal Teleconnection (EOT) analysis, and found a robust connection from observational datasets in agreement with previous studies, while confirming that GCM outputs are still not reliable for capturing this pertinent rainfall variability due to ENSO. While the strength of this relationship is not homogeneous among the region, and the patterns of vegetation response presented opposite characteristics in the northern and southern areas, ENSO influence on East African vegetation and in return its carbon and hydrological fluxes was apparent. The simulated vegetation responses showed non-negligible differences under climate with and without stronger ENSO signal in relevance to mitigation efforts for future climate change. We conclude that the future vegetation would be different from what is simulated under these climate model outputs lacking accurate ENSO contribution to the degree of ignoring the ENSO influence altogether. Comparably with findings from previous studies linking vegetation-climate interactions, we discussed the importance of accounting for this influence which can bring further environmental stress to East Africa. Overall, our results highlight that more robust projections on coupled atmosphere-ocean teleconnections can help reducing large uncertainties of the future magnitude and sign of carbon sink provided by terrestrial ecosystems by improving our understanding on the vegetation response.

*Code availability.*  All the R code used in this study can be found at github.com/istfer/ENSOpaper

## Appendix A

## A1    A1

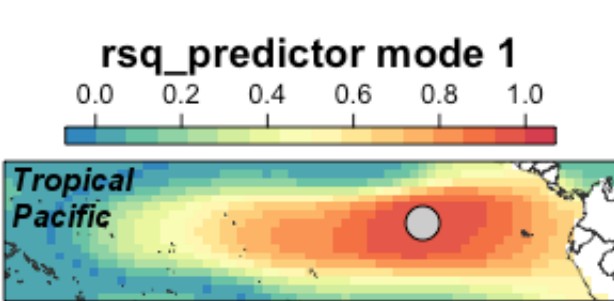

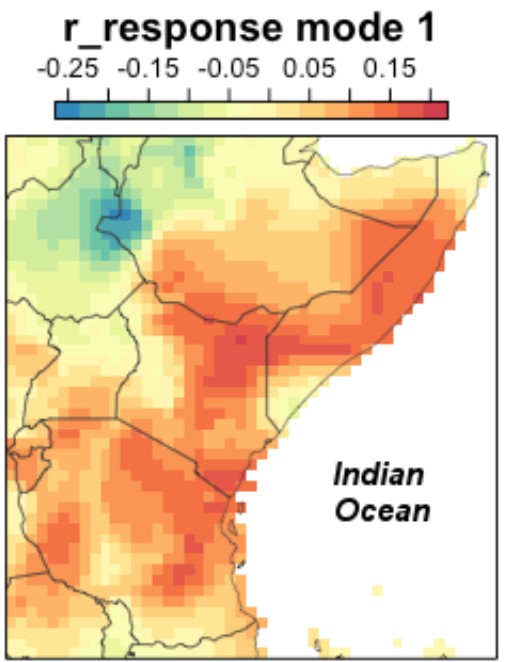

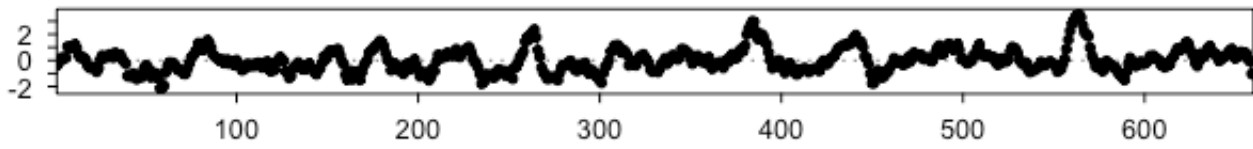

ii

**Figure A1.** Coupled ocean-atmosphere teleconnection between Pacific Sea Surface Temperatures and East African Rainfall retrieved from historical observations. (Upper Left) The coefficients of determination for the predictor field highlights that the Nino-3.4 region explains the variance in the response domain the most. (Upper Right) Correlation coefficients of the each pixel of the East Africa (response) domain shows that spatially the coastal parts and a north-western area is being explained by the predictor field. (Bottom panel) Time series of Tropical Pacific SST anomalies at the base point (the gray circle in the upper left panel) of the first mode as ENSO signal.

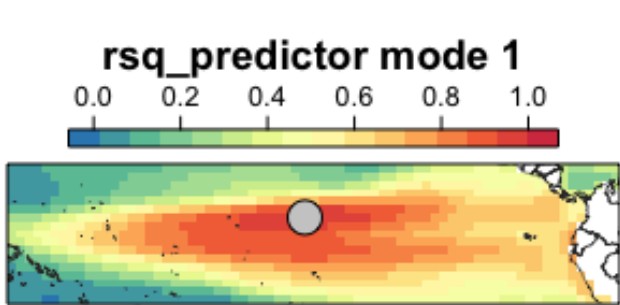

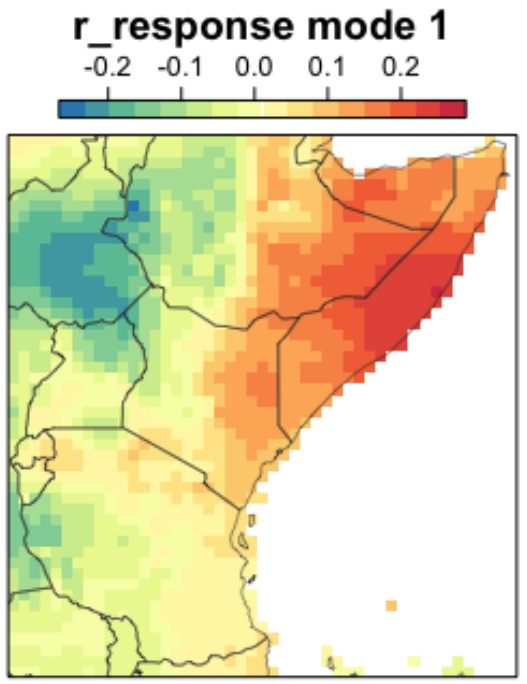

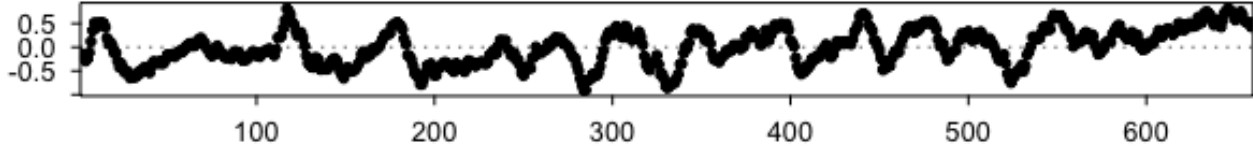

ii

**Figure A2.** EOT Analysis for the historical period from the GCM simulations. Panels as explained in Figure A1: (Left) The coefficients of determination for the predictor field. (Right) Correlation coefficients of the each pixel of the East Africa (response) domain. (Bottom) Time series at the base point of the mode.

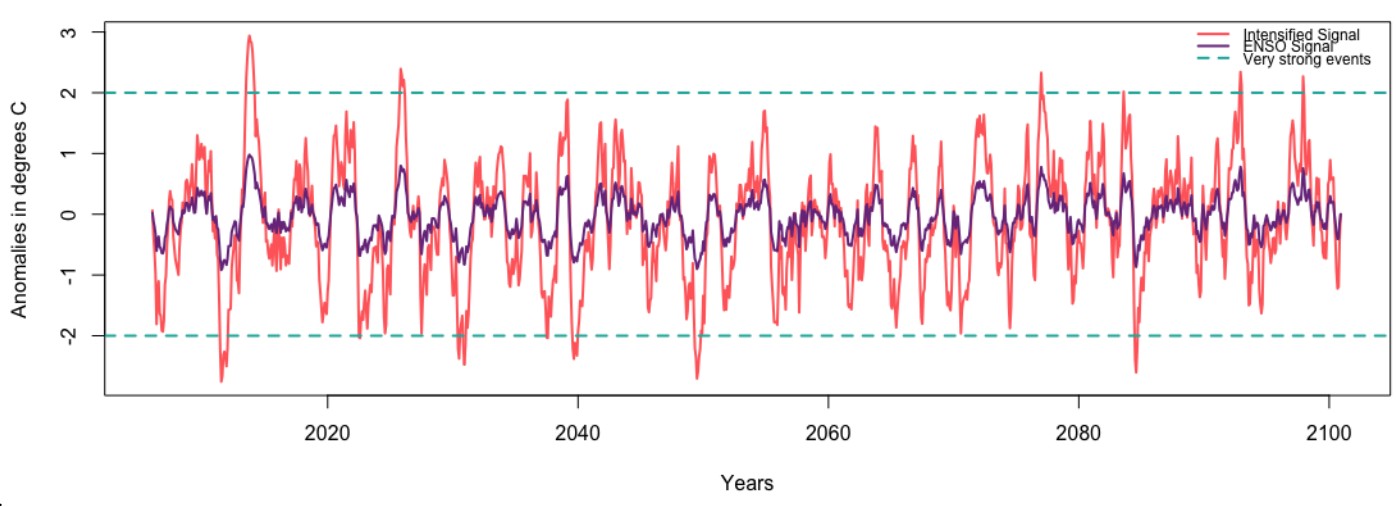

ii

**Figure A3.** Intensified ENSO signal. Purple line: Future ENSO signal retrieved from GCM outputs for 2006-2100 period. Red Line: Intensified signal such that anomalies peak as strong as recorded amplitudes ($\pm$ 2.0 $^{\circ}$C). Dashed line marks the very strong ENSO event threshold.

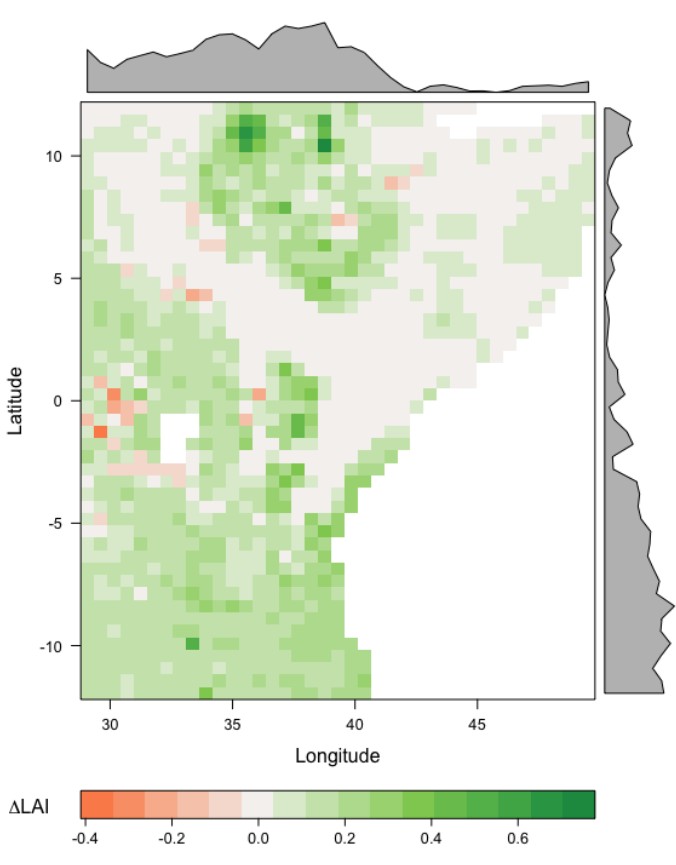

ii

**Figure A4.** Simulated woody vegetation Leaf Area Index (LAI) differences under future climate scenario RCP8.5 (without any manipulation to the ENSO signal) and present-day (PD). (ΔLAI = RCP8.5 - PD).

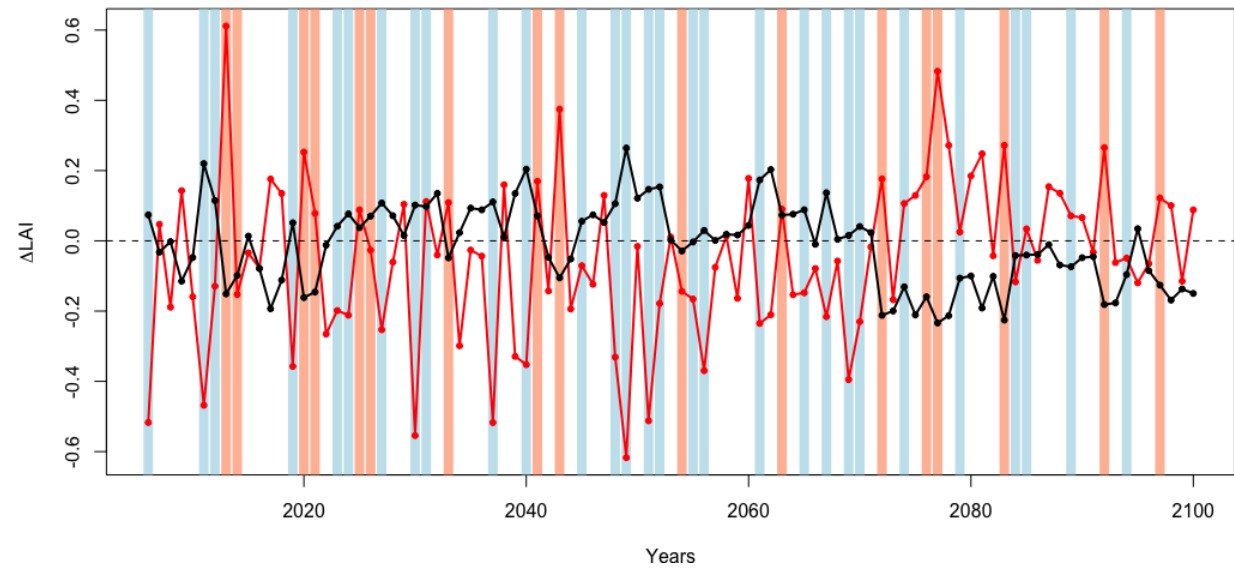

ii

**Figure A5.** Temporal differences in LAI according to future simulations with and without intensified ENSO contribution (∆ = With Intensification - Without Intensification). Black line: Northern transect, Red line: southern transect. Vertical blue lines: All moderate (< -1.0 °C) La Niña years identified for the future period (2006-2100), Vertical pink lines: Moderate (> 1.0 °C) El Niño years.

## A1  A2

CCCma-CanESM2: Canadian Centre for Climate Modelling and Analysis - The second generation Canadian Earth System Model (Flato et al., 2000)

CERFACS CNRM-CM5: Centre Européen de Recherche et de Formation Avanciée, Centre National de Recherches Météorologiques, Climate Model 5 (Voldoire et al., 2013)

IPSL CM5A-MR: Institut Pierre Simon Laplace Climate Model 5A Medium Resolution (Hourdin et al., 2013)

QCCCE CSIRO Mk3-6-0: Queensland Climate Change Centre of Excellence, Commonwealth Scientific and Industrial Research Organization, Mark 3.6 (Collier et al., 2013)

ICHEC EC-EARTH: Irish Centre for High End Computing, EC-Earth (Sterl et al., 2012)

MIROC5: Atmosphere and Ocean Research Institute (The University of Tokyo), National Institute for Environmental Studies, and Japan Agency for Marine-Earth Science and Technology, Model for Interdisciplinary Research on Climate (Watanabe et al., 2010)

MPI-M ESM-LR: Max Planck Institute for Meteorology, Earth System Model, Low Resolution (Giorgetta et al., 2013)

NCC NorESM1-M: Norwegian Climate Centre, Norwegian Earth System Model (Bentsen et al., 2013)

NOAA GFDL-ESM2M: National Oceanic and Atmospheric Administration, Geophysical Fluid Dynamics Laboratory (Dunne et al., 2012)

## A2 References

Bentsen, M., et al. (2013): The Norwegian Earth System Model, NorESM1-M - Part 1: Description and basic evaluation of the physical climate, Geosci. Model Dev., 6, 687-720, doi:10.5194/gmd-6-687-2013

Collier, M. et al., 2013, Ocean circulation response to anthropogenic-aerosol and greenhourse gas forcing in the CSIRO-Mk3.6 coupled climate model, Australian Meteorological and Oceanographic Journal, 63, 27-39

Dunne, J.P, et al., 2012, GFDL's ESM2 Global Coupled Climate-Carbon Earth System Models, American Meteorological Society, http://dx.doi.org/10.1175/JCLI-D-11-00560.1

Flato, G. M. et al., 2000. "The Canadian Centre for Climate Modeling and Analysis global coupled model and its climate". Climate Dynamics. 16 (6): 451-467. doi:10.1007/s003820050339.

Giorgetta, M., et al. 2013, Climate change from 1850 to 2100 in MPI-ESM simulations for the coupled model intercomparison project phase 5, J. Adv. Model. Earth Syst., doi:10.1002/jame.20038

5    Hourdin, F., Foujols, MA., Codron, F. et al., 2013, Clim Dyn, 40: 2167. doi:10.1007/s00382-012-1411-3

Sterl, A., Bintanja, R., Brodeau, L. et al., 2012, Clim Dyn, 39: 2631. doi:10.1007/s00382-011-1239-2

Voldoire, A., Sanchez-Gomez, E., Salas y Mélia, D. et al. Clim Dyn (2013) 40: 2091. doi:10.1007/s00382-011-1259-y

Watanabe, M., et al., 2010, Improved Climate Simulation by MIROC5: Mean States, Variability, and Climate Sensitivity, American Meteorological Society, http://dx.doi.org/10.1175/2010JCLI3679.1

*Competing interests.*   The authors declare that they have no conflict of interest.

*Acknowledgements.*   IF was funded by DAAD, grants to F. J. and German Research Foundation (DFG) Graduate School GRK1364 program (Shaping Earth's Surface in a Variable Environment - Interactions between tectonics, climate and biosphere in the African-Asian monsoonal region). FJ and BT acknowledge the support by the BMBF in the framework of the OPTIMASS project (01LL1302A and 01LL1302B). We thank Plant Ecology and Nature Conservation Group of Potsdam University for the inspiring discussions, and Dr. Appelhans for helpful discussions on the EOT method. We are grateful for the Biogeosciences' editor and the two anonymous reviewers for their comments and

suggestions that helped us improve this manuscript to a great extent.

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
