# Peer review of "Influence of El Niño-Southern Oscillation regimes on East African vegetation and its future implications under RCP8.5 warming scenario"

_Biogeosciences, 2017_

## Referee Comment (RC1) · Anonymous Referee #1 · 2 May 2017

General Comments:

This study aims to quantify the ENSO contribution to the East African rainfall variability and examine vegetation response to future rainfall variability as influenced by projected intensified ENSO. The topic is quite interesting and the manuscript is generally well-organized. It seems that authors deemed that ENSO will be intensified under the RCP8.5, high emissions scenario. So by contrast, the ENSO should be relatively weakened under the low emissions scenarios, such as RCP 2.6. In this case, comparing simulated results driven by two scenarios climate outputs should also provide useful information about vegetation response to intensified ENSO. I understand this would result in much more workload for model simulation, however, the authors are expected to explain why they only used climate outputs under the RCP8.5 scenario. Also, more quantitative results are expected in abstract, especially about projected vegetation response to ENSO. LPJ-GUESS is a dynamic vegetation model, but authors did not show any results about changes in vegetation distribution. I would be curious if vegetation distribution would change a lot under the RCP8.5 in this region. Other specific comments are below:

Specific comments: 1. Abstract, P2, L33, please specify what this study simulated, carbon and water fluxes or vegetation distribution? 2. P58-61 and P105-109, Authors kind of repeat research objectives in two places; please reorganize them accordingly. 3. Results section: need a separate sub-section to present future results. 4. Besides spatial patterns, results of temporal variations in carbon and water fluxes as influenced by ENSO are expected in results section. 5. Please have a paragraph or sub-section to identify uncertainties involved in this study.

---

## Short Comment (SC1) · 27 May 2017

I thank the reviewer for the encouraging comments and constructive suggestions.

I agree with the reviewer that the ENSO behaviour would be different under lower emission scenarios. We chose RCP8.5 only, because with a high emissions scenario such as the RCP8. 5, we intend cover the broadest range of potential changes from present to future. Furthermore, current trajectory points beyond the RCP8.5 scenario as we briefly mention in the methods, which explains why some of the recent high-profile publications also only used the RCP8.5 scenario: Burrows et al. 2014; Yoon et al. 2015; Milly et al. 2016

Following the reviewer's suggestion, we will add further discussion in the text and address the rest of the comments as soon as possible.

Papers cited above:

Burrows MT, Schoeman DS, Richardson AJ et al. (2014) Geographical limits to species-range shifts are suggested by climate velocity. Nature, 507, 492-495.

Milly PCD, Dunne KA (2016) Potential evapotranspiration and continental drying. Nature Climate Change, advance online publication.

Yoon JH, Wang SY, Gillies RR, Kravitz B, Hipps L, Rasch PJ (2015) Increasing water cycle extremes in California and in relation to ENSO cycle under global warming. Nature Communications, 6, 8657.

---

## Referee Comment (RC2) · Anonymous Referee #2 · 1 Jun 2017

The Major Problem The conclusion of the paper states: There is a relationship between the East African rainfall and ENSO events in agreement with previous studies (so nothing new), and climate models (CMIP5) are not good at capturing rainfall variability due to ENSO (also not new), therefore the future vegetation would be different from what is simulated using these climate models outputs. Both of these conclusions are already known. Thus what is new in this manuscript is the projection based on CMIP5 climate models that do not capture the most important parameter – precipitation, and very probably they also not to capture properly the temperature, which are required as inputs to the LPJ-GUESS model. Therefore the authors provide the statement that the future would be different from what is simulated using these climate models outputs.

[Figure]

Why than should be the manuscript published? The manuscript can be still useful if the authors would concentrate on the model projected differences between two plausible scenarios. If we succeed in controlling CO2 emission, we may follow a path close to the RCP4.5 scenario. If we fail to control the emission it would be close to RCP8.5. I recommend considering these two scenarios, and concentrating on model projected differences between the two alternatives (RCP4.5 and RCP8.5). This will require a major revision or a re-submission, but it will significantly improve the quality of the paper. Fact that some papers were published using only RCP8.5 should not be an excuse to continue this less than the best possible practice.

Minor Points: (1) Several CMIP5 climate models were used for the presented study. How were these models selected from about 40 existing models and why? What was the criterion for the selection? If different models would be used how would be the results changed? (2) There have been several papers published recently suggesting two kinds of El Nino events (EP and CP El Nino) with the suggestion that the future global warming will produce more El Nino just of one type. Is your El Nino projection in agreement with this statement? (3) The models used for future projection should be supported by showing an agreement with the past observations (necessary but not sufficient condition). This is not a guarantee that the models will provide reliable future projections, but if models cannot agree with the past observations their use for future projections is not justified. Since the LPJ-GUESS requires the precipitation and temperature as a part of input, please show how the ensemble mean of the CMIP5 models used simulate the past precipitation and temperature of East Africa. (4) It is now 2017, why are you using only 1951-2005 as a historical period? Historical models simulations can be extended till present (e.g. 2016) using parts (2005-2016) of future RCP projections. (5) Consider how your projections confirm or contradict recent observations of widespread greening (e.g. Forzieri et al, Science 2017; Brandt et al, Nature Ecology and Evolution 2017). (6) I don't see the urgency implied in the title. Please, consider a different title.

---

## Author Comment (AC1) · 26 Jun 2017

We thank both reviewers and the associate editor for their helpful comments and suggestions. Please find below our responses.

**Reviewer 1**

**This study aims to quantify the ENSO contribution to the East African rainfall variability and examine vegetation response to future rainfall variability as influenced by projected intensified ENSO. The topic is quite interesting and the manuscript is generally well-organized.**

[Figure]

- We are grateful to the reviewer for the encouraging comments and constructive suggestions.

**It seems that authors deemed that ENSO will be intensified under the RCP8.5, high emissions scenario. So by contrast, the ENSO should be relatively weakened under the low emissions scenarios, such as RCP 2.6. In this case, comparing simulated results driven by two scenarios climate outputs should also provide useful information about vegetation response to intensified ENSO. I understand this would result in much more workload for model simulation, however, the authors are expected to explain why they only used climate outputs under the RCP8.5 scenario.**

- Our reasons to choose RCP8.5 were two folds: i) in order to capture the furthest range as presented by RCPs, and ii) current trajectory already points beyond RCP8.5 (Sanford et al., 2014).

However, we found that the ENSO signal as identified by the EOT method to be very weak in the Global Climate Model (GCM) outputs. Intensifying it to its observed strength (Figure S3), already resulted in significant changes in the ecosystem responses. In a sense, it could be argued that we did not even applied an extra intensification due to RCP8.5 (as we discuss in the manuscript, L514-524). In other words, our results show "what would it be like if the GCMs were able to capture the coupled tropical Pacific Sea Surface temperature-East African rainfall teleconnection, assuming this teleconnections would be at least as strong as historical observations". Hence, if we were to use a lesser intensification in order to represent RCP2.6 under the same methodology, it would imply we assume ENSO would be weakened relative to present-day (not relative to RCP8.5) under RCP2.6 conditions. We intensified the future ENSO signal to be at least as strong as what is being observed historically because previous paleoclimate studies (Wollf et al., 2011) suggest that a possible future warming will intensify the interannual to centennial-scale changes in ENSO-related rainfall variability in East Africa.

**Also, more quantitative results are expected in abstract, especially about projected vegetation response to ENSO. LPJ-GUESS is a dynamic vegetation model, but authors did not show any results about changes in vegetation distribution. I would be curious if vegetation distribution would change a lot under the RCP8.5 in this region.**

- In this study, as we were mainly interested in the overall ecosystem metabolism in terms of carbon and water cycle, we focused on the fluxes. Nevertheless, LPJ-GUESS being a dynamic vegetation model is also important for obtaining more realistic predictions for fluxes as biogeochemical and hydrological cycles are linked to vegetation dynamics as well as the changing climate.

In our previous simulation studies for the region with LPJ-GUESS, we mainly focused on the changes in the vegetation distribution and composition (Fer et al., 2015; 2016). From these studies it was apparent that significant changes in East African vegetation distribution would be more likely under stronger changes in the East African climate such as the wet periods during the mid-Holocene. However, changes in vegetation distributions might still be relevant for certain locations, especially in terms of tree cover. We could provide an assessment of the vegetation cover and composition in terms of Leaf Area Index/PFT and include discussion in the text.

*Specific comments:*

**1. Abstract, P2, L33, please specify what this study simulated, carbon and water fluxes or vegetation distribution?**

- We could explicitly specify that we simulated the fluxes in the corresponding line as:

Then, we simulated the ecosystem carbon and water fluxes under the historical climate without components related to ENSO teleconnections.

**2. P58-61 and P105-109, Authors kind of repeat research objectives in two places; please reorganize them accordingly.**

- We intended the last sentence of the introduction to outline the roadmap for the paper. We could rephrase it as suggested.

**3. Results section: need a separate sub-section to present future results.**

- We thank the reviewer for pointing this out, a sub-heading is necessary and will be added to the results section.

**4. Besides spatial patterns, results of temporal variations in carbon and water fluxes as influenced by ENSO are expected in results section.**

- As we conducted the study for the whole East Africa domain, and as ENSO impact would differ from site to site, each grid cell (1550 in total) would have their own temporal variation. That is why we decided to report the spatial patterns to summarize the overall response. However, following the strategy we adopted in the paper, we could provide such temporal variations in the north and south transects and discuss accordingly.

**5. Please have a paragraph or sub-section to identify uncertainties involved in this study.**

- We thank the reviewer for this remark. Indeed the parameter and driver uncertainties would be important to discuss. LPJ-GUESS has been parameterized and validated for this region before (Fer et al. 2015), and the model sensitivity to drivers were tested to a certain extent (Fer et al., 2016). We would be happy to include further discussion on uncertainties.

*Reviewer 2*

**The Major Problem The conclusion of the paper states: There is a relationship between the East African rainfall and ENSO events in agreement with previous studies (so nothing new), and climate models (CMIP5) are not good at capturing rainfall variability due to ENSO (also not new), therefore the future vegetation would be different from what is simulated using these climate models outputs. Both of these conclusions are already known. Thus what is new in this**

**manuscript is the projection based on CMIP5 climate models that do not capture the most important parameter – precipitation, and very probably they also not to capture properly the temperature, which are required as inputs to the LPJ-GUESS model. Therefore the authors provide the statement that the future would be different from what is simulated using these climate models outputs.**

- While we agree that these two findings were known to a certain extent, they were also the main motivation of this study: there is a known relationship between East African rainfall and ENSO events, and GCMs are not good at capturing this rainfall variability due to ENSO. This immediately raises the question: what is the extent of this discrepancy? What are we missing when we drive our vegetation models with these GCMs for future projections? Is it a negligible difference or does it make our forecasts unreliable? This is a crucial gap to fill in our knowledge given that these projections are often being used in decision making. Indeed, our quantification showed importance of capturing this relationship (hence, the urgency). The reason why we reiterated the "already known" findings in the conclusion as well is that we established them independently using the EOT method in this study and presented in a self-contained framework. We would be happy to re-phrase the conclusion section to highlight this context better.

**Why than should be the manuscript published? The manuscript can be still useful if the authors would concentrate on the model projected differences between two plausible scenarios. If we succeed in controlling CO2 emission, we may follow a path close to the RCP4.5 scenario. If we fail to control the emission it would be close to RCP8.5. I recommend considering these two scenarios, and concentrating on model projected differences between the two alternatives (RCP4.5 and RCP8.5). This will require a major revision or a re-submission, but it will significantly improve the quality of the paper. Fact that some papers were published using only RCP8.5 should not be an excuse to continue this less than the best possible practice.**

- We agree that the fact that other papers were published using only RCP8.5 cannot be

an excuse to do so. However, our reasoning was the other way around: As we already mentioned above, under the current trajectory the RCP8.5 seems to be the more plausible scenario amongst the alternatives. Using all or more alternatives would certainly provide further information. However, when it comes to assessment of ecosystem services and mitigation efforts, studying the full range is strategically justified as that is the extent to be considered while defining and consolidating climate change adaptation programs. Therefore, we do not find it meritless to use RCP8.5 only in these studies, including ours.

Finally, the aim of this study is to quantify the ENSO influence on East African vegetation and understand its future implications for ecosystem services. We believe this was achieved with our current analyses without necessarily needing additional scenarios: We were able to show that driving our models with future GCM outputs as they are would be as if simulating the present-day vegetation with climate data without any ENSO contribution by only bringing the future ENSO signal to its observed strength. Hence, this discrepancy would hold regardless of the scenario.

*Minor Points:*

**(1) Several CMIP5 climate models were used for the presented study. How were these models selected from about 40 existing models and why? What was the criterion for the selection? If different models would be used how would be the results changed?**

- In this study, we used the dynamically downscaled GCM outputs by the CORDEX (Coordinated Regional Climate Downscaling Experiment) program. Regional climate models (RCMs) dynamically downscale GCM output to scales better suited to end users and are useful for understanding local climate in regions that have complex topography such as East Africa (Endris et al., 2013). Therefore we were limited to CORDEX outputs which consisted of 10 models at the time. The results would of course not exactly be the same if we had used all existing CMIP5 models. That being said, we believe

the main findings and implications of the study would remain the same as we used the ensemble mean of CORDEX outputs in our study. We can add add this statement to the text as well.

**(2) There have been several papers published recently suggesting two kinds of El Nino events (EP and CP El Nino) with the suggestion that the future global warming will produce more El Nino just of one type. Is your El Nino projection in agreement with this statement?**

- This is a very good question. In our methodology, we cannot separate between two different modes of El Nino as the coupling of tropical Pacific Sea Surface temperature-East African rainfall variability emerges from the data. However, we note that while the region that explains the most variability in East African rainfall is closer to the Nino-3.4 region in our historical analysis (Figure S1), it shifts towards the Nino-4 region in the EOT analysis with GCM outputs (Figure S2) which agrees well with the statement that the intensity of CP El Nino will be increased in the future. This would be a very good discussion point for this study, we thank the reviewer for this remark.

**(3) The models used for future projection should be supported by showing an agreement with the past observations (necessary but not sufficient condition). This is not a guarantee that the models will provide reliable future projections, but if models cannot agree with the past observations their use for future projections is not justified. Since the LPJ-GUESS requires the precipitation and temperature as a part of input, please show how the ensemble mean of the CMIP5 models used simulate the past precipitation and temperature of East Africa.**

- We thank the reviewer for pointing this out, and we cordially agree that this is an important point in vegetation modeling studies. However, we already extensively tested LPJ-GUESS for historical and mid-Holocene periods in a previous study (Fer et al., 2015). Our past simulations showed good agreement with observational data for both periods and have been reported in a peer-reviewed journal which we refer the more

interested reader to on L137-138.

**(4) It is now 2017, why are you using only 1951-2005 as a historical period? Historical models simulations can be extended till present (e.g. 2016) using parts (2005-2016) of future RCP projections.**

- We were following the division as provided by CORDEX outputs, which were using 1950-2005 as historical and 2006-2100 as future period. We could have rearranged the years but we adhered to the CORDEX setting for interpretability and reproducibility reasons.

**(5) Consider how your projections confirm or contradict recent observations of widespread greening (e.g. Forzieri et al, Science 2017; Brandt et al, Nature Ecology and Evolution 2017).**

- While it is not possible for us to directly confirm or contradict findings of these studies (Forzieri et al., a global study investigating energy budget in terms of LE and H; Brandt et al., a continental study including human population growth in their analyses; neither studies linking observations to ENSO events), we believe these papers would be important to include in our discussion. Forzieri et al. remark the importance of the interplay between LAI and surface biophysics, which reinforces our findings as ENSO affects the LAI temporal variability in East Africa. Besides wet-cool (El Nino), dry-warm (La Nina) ENSO events in East Africa is a good example of their extreme conditions case. Brandt et al. report a decreasing trend in woody vegetation over East Africa for 1992-2011 associated with high human population growth. It is interesting to see that, the regions that experience woody vegetation decrease and human population increase the most are also the regions where ENSO impact is the highest. Such interconnections will enrich our discussion, we thank the reviewer for pointing these studies out.

**(6) I don't see the urgency implied in the title. Please, consider a different title.**

- We could propose two alternatives or would be happy to conform to suggestions from the editor:

Title A: Influence of El Niño-Southern Oscillation regimes on East African vegetation and its future implications

Title B: East African vegetation response under El Niño-Southern Oscillation influence show discrepancies

***Cited literature:***

Endris, H. S., Omondi, P., Jain, S., Lennard, C., Hewitson, B., Chang'a, L., Awange, J.L., Dosio, A., Ketiem, P., Nikulin, G., Panitz, H. J., Büchner, M., Stordal, F., Tazalika, L., 2013, Assessment of the performance of CORDEX regional climate models in simulating eastern Africa rainfall, Journal of Climate. 26 (21): pp. 8453-8475, http://dx.doi.org/10.1175/JCLI-D- 12-00708.1

Fer, I., Tietjen, B., Jeltsch, F., 2015. High-resolution modelling closes the gap between data and model simulations for Mid-Holocene and present-day biomes of East Africa. Palaeogeography, Palaeoclimatology, Palaeoecology, 444, 144-151. http://dx.doi.org/10.1016/j.palaeo.2015.12.001

Fer, I., Tietjen, B., Jeltsch, F., Trauth, M.H., 2016, Modeling vegetation change during Late Cenozoic uplif of the East African plateaus, Palaeogeography, Palaeoclimatology, Palaeoecology. 467,120-130. http://dx.doi.org/10.1016/j.palaeo.2016.04.007

Sanford. T., Frumhoff, P. C., Luers, A., Gulledge, J., 2014, The climate policy narrative for a dangerously warming world. Nature Climate Change, 4, 164-166, dx.doi.org/10.1038/nclimate2148

Wolff, C., Haug, G.H., Timmermann, A., Damste, J.S.S, Brauer, A., Sigman, D.M., Cane, M.A., Verschuren, D., 2011, Reduced interannual rainfall variability in East Africa during the Last Ice Age, Science, 333, 743. dx.doi.org/10.1126/science.1203724

---

## Author Response (AR2)

**Revision notes for the manuscript :** "Accounting for El Niño-Southern Oscillation influence becomes urgent for predicting future East African ecosystem responses"

We thank both reviewers and the editor very much for their comments and suggestions. We reframed our conclusion and addressed all of the referee comments. Please find our point-by-point responses below. The line numbers given are referring to the marked-up version of the revised manuscript.

**Reviewer #1**

It seems that authors deemed that ENSO will be intensified under the RCP8.5, high emissions scenario. So by contrast, the ENSO should be relatively weakened under the low emissions scenarios, such as RCP 2.6. In this case, comparing simulated results driven by two scenarios climate outputs should also provide useful information about vegetation response to intensified ENSO. I understand this would result in much more workload for model simulation, however, the authors are expected to explain why they only used climate outputs under the RCP8.5 scenario.

As both referees were critical about the use of RCP8.5 scenario only, we now extended our discussion sub-header to include scenario selection and added a detailed paragraph on its discussion for the readership. L526-542.

Also, more quantitative results are expected in abstract, especially about projected vegetation response to ENSO. LPJ-GUESS is a dynamic vegetation model, but authors did not show any results about changes in vegetation distribution. I would be curious if vegetation distribution would change a lot under the RCP8.5 in this region.

In this study, as we were mainly interested in the overall ecosystem metabolism in terms of carbon and water cycle, we focused on the fluxes. Nevertheless, LPJ-GUESS being a dynamic vegetation model is also important for obtaining more realistic predictions for fluxes as biogeochemical and hydrological cycles are linked to vegetation dynamics as well as the changing climate.

However, changes in vegetation distributions might still be relevant for certain locations, especially in terms of tree cover. Now we added a supplementary figure (Figure A4) that shows the change in simulated woody vegetation Leaf Area Index due to change in climate and added a discussion un the text L469-477 and L519-525

**Specific comments:**

1. Abstract, P2, L33, please specify what this study simulated, carbon and water fluxes or vegetation distribution?

We now explicitly specify that we simulated the fluxes in the corresponding line. L25

2. P58-61 and P105-109, Authors kind of repeat research objectives in two places; please reorganize them accordingly.

We intended the last sentence of the introduction to outline the roadmap for the paper. We now rephrased it as suggested. L88

3. Results section: need a separate sub-section to present future results.

We thank the reviewer for pointing this out, a sub-heading is necessary and is now added to the results section. L370

4. Besides spatial patterns, results of temporal variations in carbon and water fluxes as influenced by ENSO are expected in results section.

As we conducted the study for the East Africa domain, and as ENSO impact would differ from site to site, each grid cell (1550 in total) would present their own temporal variation. That is why we decided to report the spatial patterns to summarize the overall response. However, now we added a figure (Figure 5) showing such temporal variations in the north and south transects L386-390

5. Please have a paragraph or sub-section to identify uncertainties involved in this study.

We thank the reviewer for this remark. We now included such a paragraph in the manuscript L499-508

**Reviewer #2**

The Major Problem The conclusion of the paper states: There is a relationship between the East African rainfall and ENSO events in agreement with previous studies (so nothing new), and climate models (CMIP5) are not good at capturing rainfall variability due to ENSO (also not new), therefore the future vegetation would be different from what is simulated using these climate models outputs. Both of these conclusions are already known. Thus what is new in this manuscript is the projection based on CMIP5 climate models that do not capture the most important parameter – precipitation, and very probably they also not to capture properly the temperature, which are required as inputs to the LPJ-GUESS model. Therefore the authors provide the statement that the future would be different from what is simulated using these climate models outputs.

While we agree that these two findings were known to a certain extent, they were also the main motivation of this study: there is a known relationship between East African rainfall and ENSO events, and GCMs are not good at capturing this rainfall variability due to ENSO. Then it raises the very question that what is the extent of this discrepancy? What are we missing when we drive our vegetation models with these GCMs for future projections? Is it a negligible difference or does it make our forecasts unreliable? This is a crucial gap to fill in our knowledge given that these projections are often being used in decision making. Indeed, our quantification showed importance of capturing this relationship (hence, the urgency). The reason why we reiterated the "already known" findings in the conclusion as well is that we established them independently using the EOT method in this study and presented in a self-contained framework. We now re-phrased the conclusion section to highlight this context better.

Why than should be the manuscript published? The manuscript can be still useful if the authors would concentrate on the model projected differences between two plausible scenarios. If we succeed in controlling CO2 emission, we may follow a path close to the RCP4.5 scenario. If we fail to control the emission it would be close to RCP8.5. I recommend considering these two scenarios, and concentrating on model projected differences between the two alternatives (RCP4.5 and RCP8.5). This will require a major revision or a re-submission, but it will significantly improve the quality of the paper. Fact that some papers were published using only RCP8.5 should not be an excuse to continue this less than the best possible practice.

In this study we aimed to quantify the ENSO influence on East African vegetation and understand its future implications for ecosystem services. We believe, we were able to accomplish this aim with our current set of analyses as intended.

We now included further discussion in the manuscript explicitly for the scenario selection. L526-542

**Minor Points:**

(1) Several CMIP5 climate models were used for the presented study. How were these models selected from about 40 existing models and why? What was the criterion for the selection? If different models would be used how would be the results changed?

In this study, we used the dynamically downscaled GCM outputs by the CORDEX (Coordinated Regional Climate Downscaling Experiment) program. Regional climate models (RCMs) dynamically downscale GCM output to scales better suited to end users and are useful for understanding local climate in regions that have complex topography such as eastern Africa (Endris et al., 2013). Therefore we were limited to CORDEX outputs which consisted of 10 models at the time. The results would of course not exactly be the same if we had used all existing CMIP5 models. That being said, we believe the main findings and implications of the study would remain the same as we used the ensemble mean of CORDEX outputs in our study.

(2) There have been several papers published recently suggesting two kinds of El Nino events (EP and CP El Nino) with the suggestion that the future global warming will produce more El Nino just of one type. Is your El Nino projection in agreement with this statement?

We thank the reviewer for pointing this out. We now included the discussion on EP and CP ENSO in our manuscript. L551-564

(3) The models used for future projection should be supported by showing an agreement with the past observations (necessary but not sufficient condition). This is not a guarantee that the models will provide reliable future projections, but if models cannot agree with the past observations their use for future projections is not justified. Since the LPJ-GUESS requires the precipitation and temperature as a part of input, please show how the ensemble mean of the CMIP5 models used simulate the past precipitation and temperature of East Africa.

We thank the reviewer for this remark, and we cordially agree that this is an important point in vegetation modeling studies. However, we already extensively tested LPJ-GUESS for historical and mid-Holocene periods in a previous study (Fer et al., 2015). Our past simulations showed good agreement with observational data for both periods and have been reported in a peer-reviewed journal which we refer the more interested reader to on L117-119.

(4) It is now 2017, why are you using only 1951-2005 as a historical period? Historical models simulations can be extended till present (e.g. 2016) using parts (2005-2016) of future RCP projections.

We were following the division as provided by CORDEX outputs, which were using 1950-2005 as historical and 2006-2100 as future period. We could have rearranged the years but we adhered to the CORDEX setting for interpretability and reproducibility reasons. We now, explicitly state this in the text. L138

(5) Consider how your projections confirm or contradict recent observations of widespread greening (e.g. Forzieri et al, Science 2017; Brandt et al, Nature Ecology and Evolution 2017).

We thank the reviewer for these suggestions. We now included discussion regarding these two studies in the text. L469-477 and L519-525.

(6) I don't see the urgency implied in the title. Please, consider a different title.

We now changed the title to : "Influence of El Niño-Southern Oscillation regimes on East African vegetation and its future implications under RCP8.5 warming scenario"

Cited literature:

Endris, H. S., Omondi, P., Jain, S., Lennard, C., Hewitson, B., Chang'a, L., Awange, J.L., Dosio, A., Ketiem, P., Nikulin, G., Panitz, H. J., Büchner, M., Stordal, F., Tazalika, L., 2013, Assessment of the performance of

CORDEX regional climate models in simulating eastern Africa rainfall, Journal of Climate. 26 (21): pp. 8453-8475, http://dx.doi.org/10.1175/JCLI-D- 12-00708.1

Fer, I., Tietjen, B., Jeltsch, F., 2015. High-resolution modelling closes the gap between data and model simulations for Mid-Holocene and present-day biomes of East Africa. Palaeogeography, Palaeoclimatology, Palaeoecology, 444, 144-151. http://dx.doi.org/10.1016/j.palaeo.2015.12.001

Fer, I., Tietjen, B., Jeltsch, F., Trauth, M.H., 2016, Modeling vegetation change during Late Cenozoic uplif of the East African plateaus, 
[revised manuscript text omitted]

$$EA_{dns} = \text{denoise}(EA_{ds}) \qquad PAC_{dns} = \text{denoise}(PAC_{ds}) \qquad (2)$$

ii) Conduct Empirical Orthogonal Teleconnection (EOT) analysis:

$$EOT_{modes} \leftarrow EOT(EA_{dns} \sim PAC_{dns}) \qquad (3)$$

Here the $EOT_{modes}$ object can be thought as a list that stores both the time series of the modes, the reduced fields obtained after the removal of each mode, slopes and intercepts of the fields (for more details see Appelhans  et al., 2015).

iii) Calculate the difference (*Diff*) between the de-seasoned, de-noised data ($EA_{dns}$) and the rainfall behaviour without ENSO contribution from the information that is already stored in the resulting

$EOT_{modes}$ object (ENSO signal is the first mode, therefore the rainfall behaviour we are left without ENSO will be the $EA_{modes,\ rr1}$ where subscript *rr1* indicating "response residual" after the removal of the first EOT mode:

$$\text{Diff} = EA_{dns} - EA_{modes,\ rr1} \tag{4}$$

iv) If we subtract this difference from the initial raw response field ($EA_r$), we will obtain the East

African precipitation without ENSO contribution ($EA_{r,\ woENSO}$):

$$EA_{r,\ woENSO} = EA_r - \text{Diff} \tag{5}$$

v) As EOT analysis is basically a regression analysis, we can also obtain the ENSO contribution (*Diff*) from the regression equation as shown below (which will become handy when we insert back the intensified ENSO signal):

$$\text{Diff} = EOT_{modes,\ eot\_1} * EOT_{modes,ri\_1} - EOT_{modes,\ rs\_1} \tag{6}$$

Here $EOT_{modes,\ eot\_1}$, $EOT_{modes,ri\_1}$ and $EOT_{modes,\ rs\_1}$ refer to the EOT time series of the 1[st] mode (the ENSO signal), intercept of and slope of the response field calculated for the 1[st] mode (Appelhans et al., 2015).

vii) Then, it is possible to modify the future ENSO signal ($EOT_{modes,\ eot\_m}$) obtained from EOT analysis on simulation datasets, re-calculate its contribution to the East African rainfall ($Diff_{new}$) and add this amount back on the precipitation data without ENSO signal ($EA_{r,\ woENSO}$) to obtain new precipitation amounts ($EA_{r,\,new}$) due to new signal. We can later use this $EA_{r,\,new}$ as the future precipitation input to our vegetation model to drive future simulations.

$$\text{Diff}_{new} = \text{EOT}_{modes,\,eot\_m} * \text{EOT}_{modes,ri\_1} - \text{EOT}_{modes,\,rs\_1} \tag{7}$$

[revised manuscript text omitted]